# Pathophysiological Role and Potential Therapeutic Exploitation of Exosomes in Ovarian Cancer

**DOI:** 10.3390/cells9040814

**Published:** 2020-03-27

**Authors:** Aasa Shimizu, Kenjiro Sawada, Tadashi Kimura

**Affiliations:** Department of Obstetrics and Gynecology, Osaka University Graduate School of Medicine, 2-2, Yamadaoka, Suita, Osaka 5650871, Japan; aasashimizu@gyne.med.osaka-u.ac.jp (A.S.); tadashi@gyne.med.osaka-u.ac.jp (T.K.)

**Keywords:** exosome, ovarian cancer, peritoneal dissemination, exosome-based therapy, drug delivery

## Abstract

Exosomes are extracellular vesicles involved in several biological and pathological molecules and can carry many bioactive materials to target cells. They work as important mediators of cell-cell communication and play essential roles in many diseases, especially in cancer. Ovarian cancer is one of the most common gynecological malignancies. Most patients are diagnosed at advanced stages involving widespread peritoneal dissemination, resulting in poor prognosis. Emerging evidence has shown that exosomes play vital roles throughout the progression of ovarian cancer. Moreover, the development of engineered exosome-based therapeutic applications— including drug delivery systems, biomolecular targets and immune therapy—has increased drastically. Herein, we review the functional features of exosomes in ovarian cancer progression and the therapeutic application potential of exosomes as novel cancer treatments.

## 1. Introduction

Almost all cells release various types of extracellular vesicles (EVs), including exosomes, microvesicles and apoptotic bodies [1]. EVs vary in size, properties and secretion pathways, depending on the originator cells. Exosomes are small EVs, which are approximately 100 nm in diameter and are released into the extracellular microenvironment to transfer their components, such as messenger RNAs (mRNAs), micro RNAs (miRNAs), DNAs and proteins into their target cells. Microvesicles are larger than exosomes, approximately 100–1000 nm in diameter and are composed of the lipid components of the plasma membrane [2]. Apoptotic bodies, which are less than 5 μm in size, are released from apoptotic cells. In recent years, an increasing number of physiological functions of EVs have been reported [3] and EVs have increasingly been found to play crucial roles in many biological processes, including intercellular communication, immune function, development and differentiation of stem cells, cell signaling, carcinogenesis and tumor prognosis [4]. The existence of EVs was first reported in 1983. Transferrin receptors of reticulocytes were released via small vesicles of about 50 nm in size during in vitro maturation [5]. Around that time, such small vesicles were considered as carriers meant to dispose unnecessary intracellular materials. However, in 2007, Valadi et al. showed that EVs contain both mRNA and miRNA that can be transferred to another cell and become functional in the new location [6]. Following this report, EVs received significant attention from researchers due to its potential as a cell-cell communicator through the transfer of oligonucleotides, bioactive proteins and lipids. These loading biomolecules work on recipient cells and modify protein production and gene expression [7,8,9], which supports cancer cell progression and metastasis. In recent times, EVs have been reported to play a pivotal role in cancer prognosis [10].

Ovarian cancer is the most lethal human gynecological malignancy and accounts for approximately 5% of cancer deaths among women. When diagnosed at an early stage, its prognosis is good and the five-year relative survival rate exceeds 90%. However, more than two-thirds of ovarian cancer patients are unfortunately diagnosed at advanced stages, which involve extensive peritoneal dissemination with massive ascites and acquired chemo-resistance during the treatment course [11]. Indeed, patients with massive peritoneal dissemination throughout the abdominal cavity are usually incurable in more than 20% of the cases. Although a variety of novel drugs, including molecular target drugs, immune checkpoint inhibitors and pharmacological inhibitors have been developed and applied for clinical trials, the clinical cure rate in patients with advanced-stage ovarian cancer has not substantially improved [12,13,14]. To overcome this lethal disease, the discovery of other novel therapeutic approaches is necessary.

Recently, as promising therapeutic interventions for ovarian cancer, EVs as represented by exosomes have received great attention due to their molecular dynamics. Taking these biochemical advantages into consideration, elucidation of the relationship between ovarian cancer and exosomes might help us identify the underlying mechanism of peritoneal dissemination and develop a novel therapeutic strategy. Furthermore, owing to the high specific organotropism and stability in the bloodstream, exosomes are expected to be novel nano-based drug delivery carriers. Dynamic developments in the engineering of exosome-based treatment might lead to the development of novel molecular targeted therapies. In this review, we highlight the latest findings that describe the role of exosomes in ovarian cancer pathogenesis and the potential of engineering exosomes for therapeutic application in cancer therapies.

## 2. Overview of Exosomes

EVs are small endogenous phospholipid bilayer vesicles with typical diameters of about 40–1000 nm, which are secreted from a variety of cell types. EVs can be isolated from cell culture supernatants or body fluids. Based on differences in biosynthesis and size, EVs are divided into three subgroups: membrane-shedding EVs (microparticles), multivesicular body-derived EVs (exosomes) and apoptosis-derived EVs (apoptotic bodies) [4,15]. Microparticles (also known as microvesicles) are medium-sized vesicles (50–3000 nm); apoptotic bodies are lager vesicles (800–5000 nm), whereas, exosomes are comparatively smaller-sized vesicles (40–150 nm). However, it is not possible to separate these three vesicles precisely since no consensus on a “gold-standard” method to isolate and purify these microvesicles currently exists. Therefore, the International Society for Extracellular Vesicles (ISEV) recommends that these vesicles are called EVs collectively [16]. Since the majority of previous studies have focused on the roles of exosomes, we focus on the mechanisms of exosomes in this review manuscript.

Previous research on exosomes has mainly focused on their varied biological features. First, exosomes transport many biological molecules such as mRNA, miRNA [6], non-cording RNAs [17], DNA, lipid and protein [18]. The detailed information on the biological molecules in exosomes are available at exosome databases such as Vesiclepedia (http://microvesicles.org), EVpedia (https://omictools.com/evpedia-tool) and ExoCarta (http://www.exocarta.org). Second, these bioactive molecules encased by exosomes are delivered in a considerably stable condition. In general, nucleic acid is immediately degraded by endonuclease present in the extracellular space, physiological fluids and immune cells. However, exosomes protect these molecules from degradations caused by endonuclease [19]. CD47—one of the integral associated proteins referred to as the “do not eat me” signal—preserves cells from phagocytosis by macrophage and monocytes; the expression of CD47 on the surface of exosomes serves to increase the time of exosome circulation in the blood and preserve the biological activity of its cargo. Furthermore, exosomes have a natural targeting capability to unravel pathological mechanisms and lead to the development of new therapies [20]. Exosomes secreted from donor cells can transport their biological components into recipient cells. Also, exosomes can cross specific physical barriers, such as the peritoneal membrane and the blood-brain barrier [21,22]. Once exosomes are absorbed by the recipient cells, stored lipids, proteins, mRNAs, miRNAs, other molecules can affect the cellular function and phenotype of recipient cells through the regulation of key signaling cascade pathways, key enzyme reactions, cellular homeostasis or other mechanisms. Given that exosomes have these properties as an efficient intracellular communication tool, exosomes are considered as promising candidates for therapeutic application; a good understanding of exosomes would help unravel the mechanism of carcinogenesis and tumor metastasis.

## 3. Exosomes in Ovarian Cancer

Exosomes mediate intercellular information exchange. In the field of cancer research, exosomes have been shown to transport carcinogenesis and cancer progression-related oncogenic materials and proteins to cells in the local and distant microenvironment [23]. Exosomes orchestrate multiple systemic pathophysiological pathways, including local progression, angiogenesis and reprogramming of the microenvironment to promote cancer cell progression and support pre-metastatic niche formation, subsequent metastasis and drug resistance [24]. In this section, the relationship between exosomes and ovarian cancer has been described.

### 3.1. Epidemiology of Ovarian Cancer

Ovarian cancer is known as the “silent killer” because this type of cancer spreads broadly without the occurrence of any symptoms [25]. Ovarian cancer is one of the most common cancers and the leading cause of deaths from gynecological malignancies in women all over the world. Global cancer statistics from 185 countries indicated 295,414 new cancer cases and 184,799 deaths from this disease in 2018 [26]. More than 70% of patients with ovarian cancer are diagnosed at advanced stages with large volume ascites and peritoneal dissemination. Patients with tumors that have spread throughout the abdominal cavity can be cured in less than 20% of cases due to disease recurrence as a result of resistance to platinum-based chemotherapy acquisition [27]. Once platinum-resistant recurrences occur, only a few successful therapeutic options exist. Further chemotherapeutic agents yield responses in the range of 15–20% and a median progression-free survival rate of approximately 4 months [28]. Considering its high degree of lethality, there is an urgent need to identify the underlying cause of the disease and novel therapeutic targets. Exosomes can be promising molecular targets, since recent evidence has revealed that ovarian cancer-derived exosomes are involved in the process of ovarian cancer progression, especially in peritoneal dissemination and the formation of pre-metastatic niche [29].

### 3.2. Roles of Exosomes During Ovarian Cancer Progression

Advanced ovarian cancer is characterized by numerous peritoneal disseminations with massive ascites; a variety of cell-cell communications dynamically affects the progression of this disease. Ovarian cancer-derived exosomes can be detected from various body fluids such as ascites, blood serum and urine. Proteomic analysis revealed that these molecules include epithelial cell surface antigen (EpCAM), proliferation cell nuclear antigen (PCNA), tubular beta-3 chain (TUBB3), epidermal growth factor receptor (EGFR), apolipoprotein E (APOE), claudin 3 (CLDN3), fatty acid synthase (FASN) and ERBB2 [30], each of which plays a crucial role in tumorigenesis or cancer progression. Table 1 and Figure 1 summarize the content of ovarian cancer-derived exosomes that promote ovarian cancer progression by affecting their microenvironment, which is composed of fibroblast, mesothelial cell, immune cells and actual tumor cells.

Ovarian cancer-derived exosomes convert fibroblasts, which are essential for normal tissue homeostasis and function, into cancer-associated fibroblasts (CAFs) [48]. Further, CAF-derived exosomes contain TGFβ1; exosomal TGFβ1 enhanced the migration and invasion ability of ovarian cancer cells and the promotion of epithelial-mesenchymal transition (EMT) by activating the SMAD signaling [37]. Ovarian cancer derived-exosomes facilitate a variety of immune-escape systems that assist ovarian cancer progression. For instance, they induce apoptosis of dendritic cells and inhibit the function of T-cells and natural killer (NK) cells. Ovarian cancer derived exosomes can enhance the production of IL-6 in monocytes through toll-like receptor (TLR) activation. Thereafter, IL-6 activates the signal transducer and activator of transcription 3 (STAT3) pathway in immune cells, stromal cells and tumor cells, which supports immune escape of cancer cells [49]. As tumors grow, their metabolic demands outgrow blood supply, thus, they become increasingly hypoxic. In response to low oxygen concentrations, ovarian cancer increases the secretion of exosomes and promotes vessel recruitment. Activating transcription factor 2 (ATF2), metastasis-associated protein 1 (MTA1) and CD147 are included in ovarian cancer-derived exosomes that induce angiogenesis and vascular permeability [31,50]. Metastasis-associated lung adenocarcinoma transcript 1 (MALAT1), one of the exosomal non-coding RNA, also promoted angiogenesis [32]. Regarding miRNA, ovarian cancer-derived exosomes deliver miRNAs (miR-21-3p, miR-125b-5p, miR-181d-5p and miR940) to macrophages and elicit the polarization of tumor-associated macrophages (TAMs) by regulating the suppressor of cytokine signaling (SOCS)4/5/signal transducer and activator of transcription 3 (STAT3) pathways [44,45]. However, miR-223 in TAM-derived exosomes reduce the sensitivity of ovarian cancer cells to cisplatin [38].

Ovarian cancer cells release exosomes to remodel the mesothelial layer for enhanced peritoneal dissemination. CD44 in ovarian cancer-derived exosomes are transferred to peritoneal mesothelial cells (PMCs) in the initial step of peritoneal dissemination. The increased CD44 expression in PMCs promote cancer invasion by inducing PMCs to secrete matrix metalloproteinase (MMP) 9 and by destroying the mesothelial barrier for improved cancer cell invasion [47]. Exosomal miR-99a-5p upregulate fibronectin and vitronectin expression in PMCs and enhance cancer cell invasion [46]. Exosomal MMP1 mRNAs loaded in ovarian cancer-derived exosomes induce apoptotic changes in PMCs and disrupt the peritoneal mesothelial barrier, promoting the invasion of ovarian cancer cells [21].

Patients with advanced ovarian cancer, especially high-grade serous ovarian carcinoma, show large volumes of malignant ascites and exosomes in malignant ascites are also reported to promote tumor progression. A variety of studies have reported that malignant ascites contain a lot of tumor-promoting molecules, including exosomes [51] and inflammatory cytokines such as interleukin (IL)-6, IL-8, IL-10, hepatocyte growth factor (HGF), TGFβ [29]. In addition, malignant ascites-derived exosomes promote tumor progression. Malignant ascites-derived exosomes contain soluble L1, membrane-type matrix metalloproteinase (MT1-MMP), MMP-2 and urokinase-type plasminogen activator (uPA), which promote cancer migration [35,36]. Malignant ascites-derived exosomes also contribute to angiogenesis. Soluble E-cadherin in malignant ascites-derived exosomes are involved in this process [33]. Another role of ascites-derived exosomes is to serve as the immunosuppressor. Arginase-1 (ARG1)-carrying exosomes accelerate ovarian cancer growth by suppressing T-cells Phosphatidylserine in ascites inhibits T-cell activity and Fas ligand induces the apoptosis of T-cells [41,42,43] Labani-Motlagh et al. reported that phosphatidylserine, natural-killer receptor group 2, member D (NKG2D) and DNX accessory molecule-1 (DNAM-1) ligands in ascites-derived exosomes inhibited NK cell activity, resulting in immune suppression [40]. As described above, exosomes contribute to multiple steps during invasive processes and to early steps in peritoneal dissemination and metastasis.

## 4. Therapeutic Applications of Exosomes in Ovarian Cancer

Accumulating evidence has shown the utility of exosomes as carriers for drug delivery; there is an increasing expectation of the establishment of therapeutic exosome-based strategies. Exosomes can serve as efficient therapeutic modalities as well as therapeutic targets for cancer treatment [52,53]. In this chapter, we introduce recent studies on the therapeutic potential of exosomes and discuss further perspectives on their potential as regards to ovarian cancer therapy.

### 4.1. Exosomes as Drug Delivery Vehicles for Cancer Treatment

An appropriate drug delivery system (DDS) has been eagerly demanded for cancer treatment as it can not only enhance the efficacy of the loaded drug, but also reduce medical expenses, resulting in a decreased burden on patients. Among a variety of possible candidates, there are high expectations for exosomes as a promising tool of DDS since they have the potential to overcome current pharmacokinetic problems. Liposomes and polymeric nanoparticles have been the most common engineered systems for DDS for cancer therapy. Compared with these nanovectors, exosomes can avoid the endosomal and lysosomal pathways and deliver their cargoes directly into cytoplasm of recipient cells [54]. Further, exosomes are natural products of the body and therefore they exhibit a high level of stability in the circulation and a low immune response. Previous studies have shown that exosomes are specifically taken up by cancer cells rather than normal cells through endocytosis [55], although the underlying mechanisms have not been completely understood. Goh et al. reported that exosomes present the particular tetraspanins and integrins on their surface, which lead to fulfil the important role of uptake and targeting ability [56]. As a result, they can deliver their cargo to specific targets over long distances due to their high specificity for the target cells. Furthermore, exosomes have a hydrophilic core, therefore, they can load water-soluble drugs. Considering these properties, their therapeutic applications as drug delivery systems remain attractive research areas [55,57]. Specifically, advances in nanotechnology can enable the encapsulation of therapeutic agents such as small molecules, miRNA and small interfering RNA (siRNA) into exosomes. In general, there are two methods to using exosomes as a drug delivery system: exogenous method (cargo loading after EV isolation) and endogenous method (cargo loading in the steps of EVs formation). Each loading strategy, with associated advantages and limitations are summarized in Table 2.

### 4.2. Exogenous Method of Cargo Loading Into Exosomes

For therapeutic applications, the development of suitable methods for loading into exosomes is essential. Typical methods include incubation, electroporation and sonication. Incubation is the simplest way to incorporate cargo into exosomes. Hydrophobic drugs interact with the vesicles lipid layers; the drugs diffuse into the exosome cavity along the concentration gradient. Electroporation can be used for loading bioactive products into exosomes. An electrical field disturbs the phospholipid bilayer of vesicles and creates small pores in their membrane, thus allowing the passage of these bioactive products into the vesicles. Thereafter, the integrity of the vesicle membrane can be naturally recovered, resulting in the formation of drug-loaded vesicles. Sonication is also able to load biological cargo into exosomes. Exosomes derived from donor cells are mixed with drugs and subsequently sonicated by a probe sonicator, which allows the drug to flow into the exosomes due to the sonication-induced deformation of their membrane [66]; the reformation of exosome membranes enables the transit of drugs across relatively dense lipid bilayers. In this section, we focus on each cargo and refer to exogenous methods of loading them.

#### 4.2.1. Chemotherapeutic Drugs

Saari et al. demonstrated paclitaxel loading just by coincubation and showed that paclitaxel loaded in exosomes had enhanced cytotoxic effects [58]. Although this method involves lesser damage to the membrane structure of exosomes, its low loading capacity is a critical bottleneck. Curcumin is known to cause lipid rearrangement and changes in lipid fluidity of the cell membrane. Zhuang et al. suggested that curcumin may facilitate the entry of molecules into the lumen of exosomes [67]. Kim et al. showed that macrophage-derived exosomes carrying paclitaxel exhibited increased cytotoxicity to drug-resistant Mandin-Daryb canine kidney cells (more than 50 times the cytotoxicity of regular paclitaxel treatment). They also demonstrated that intravenous treatment with macrophage-derived exosomes specifically delivered paclitaxel to cancer cells in a murine Lewis lung cancer model [59]. Yanhua et al. purified exosomes from mouse immature dendritic cells and fused them with αv integrin-specific iRGW peptide. They loaded doxorubicin into these exosomes via electroporation and found that doxorubicin was efficiently delivered to αv integrin-positive breast cancer, which led to the inhibition of tumor growth without overt toxicity [60]. Kim et al. tested sonication as a method for loading paclitaxel into macrophage-derived exosomes. This method resulted in a high loading efficiency accompanied by sustained drug release, while it did not significantly affect the protein or lipid contents of the exosomes [57]. These data strongly suggest that exosome-encapsulated chemotherapeutic drugs can exert robust cytotoxic effects to cancer cells without increasing apparent adverse events, indicating they can serve as a promising novel delivery platform for the treatment of ovarian cancer which is resistant to the standard chemotherapy.

#### 4.2.2. siRNA

siRNA is a powerful tool for achieving specific gene silencing through RNA interference. Although several clinical trials based on siRNA therapeutics have recorded progress, delivering a therapeutic nucleotide to target tissues remains challenging. RNA-based therapeutics exhibited poor pharmacological properties, such as off-targeting, low serum stability and innate immune responses. Therefore, it is essential to develop novel delivery systems for siRNA and miRNA molecules that can target specific cells and tissues [68]. Exosomes can also be used as carriers that can deliver these endogenous biological cargoes [6,69]. Since siRNA molecules are relatively large and cannot be incorporated into exosomes easily due to their hydrophilic nature, the electroporation method—which is superior to the chemical transfection method—has been often used for incorporation [70]. Shtam et al. delivered siRNAs against RAD 51 and RAD 52 into exomes derived from HeLa and HT1080 cells. They successfully transported these engineered exosomes to cancer cells, resulting in effective gene silencing in recipient cells [61]. Kamerkaer et al. engineered exosomes derived from normal fibroblast-like mesenchymal cells to carry siRNAs specific to oncogenic KRASG12D. Compared to liposomes, engineered exosomes targeting KRAS showed an enhanced CD47-dependent efficacy; they evaded phagocytosis, which resulted in increasing their half-life in the circulation [55]. Further, Mayela et al. analyzed the shelf life, biodistribution, toxicology profile and efficacy of the engineered exosomes derived from bone marrow mesenchymal stem/stromal cells (MSCs), which were loaded with siRNA targeting KRASG12D in combination with chemotherapy. These engineered exosomes were strongly accumulated in tumors; the combination therapy of exosomes and gemcitabine significantly enhanced the response to gemcitabine. Also, chemotherapy-mediated debulking of large tumors potentially enhanced the efficacy of siRNA engineered exosomes [62]. In lieu of these, electroporation appears to be a favorable loading method in a clinical setting since the parameters of such settings can be controlled easily and are reproducible. However, Kooijmans et al. reported that electroporation could induce the aggregation of exosome particles and siRNAs, resulting in a 0.05–25% efficiency of siRNA retention in exosomes after electroporation [71]; this describes the potential weakness of this method. Sonication was also used to load small RNAs (siRNA and miRNA) into exosomes, although the sonication method may also induce cargo aggregation or degradation. Lamichhane et al. reported that exosomes loaded with siRNA ultimately reduced the expression of HER2, an oncogenic receptor of tyrosine kinase that critically mediates breast cancer development and progression [63].

#### 4.2.3. miRNA

miRNAs, small (22–25 nucleotides in length) non-cording RNAs, inhibit gene expression post-transcriptionally by binding to their 3’ untranslated region, leading to the suppression of protein expression or the cleavage of their target mRNAs. More than 50% of miRNA target genes are located in cancer-associated genomic regions or fragile sites, suggesting that miRNAs are deeply involved in cancer pathogenesis [72]. Considering the pivotal roles of miRNAs in carcinogenesis and cancer progression, they hold promise as therapeutic strategies against ovarian cancer. Although several miRNA therapies have been attempted in preclinical trials and many promising results have been reported, the outcomes of a few translational clinical trials for miRNA therapy have been disappointing [73]. Several bottlenecks exist for the future success of miRNA replacement therapy; one is that a single miRNA is insufficient to achieve clinical purposes since there should be a variety of cancer-related miRNAs and multiple targets; another is the rapid degradation of oligonucleotides by endonucleases within the blood. Further, since each miRNA regulates hundreds of gene expression, therapeutic miRNAs might induce unwanted effects when they are introduced into normal cells [72]. Therefore, it is obvious that improving miRNA delivery systems by developing more specific carriers and manipulating epigenetic factors that enhance miRNA function would be pivotal: exosomes are promising candidates to deliver these miRNAs to target cells. Bryniarski et al. showed that miR-150 loaded exosomes decreased endothelial cell migration and mediated suppression of effector T-cells [74]. Zhang et al. demonstrated miRNA loading, following a calcium chloride-mediated transfection or electroporation in EVs for in vitro and in vivo delivery [64]. Margherita et al. delivered miR-31 and miR-451a into exosomes derived from the plasma using electroporation and successfully transported these engineered exosomes to cancer cells, resulting in gene silencing and an enhanced apoptosis of the hepatocellular carcinoma cells [65].

### 4.3. Endogenous Method for Cargo Loading Into Exosomes

A fascinating approach to the development of tumor-tropic exosomes is the direct modification of parent cells (e.g., through genetic engineering or medication with cytotoxic drugs). Through the formation and release of exosomes, tumoricidal contents or cytosolic milieu are packaged into ensuing exosomes [75]. Transfection is a common and efficient method for loading therapeutic proteins or oligonucleotides into donor cells. Thereafter, a certain gene product would be packaged into the exosome lumen. Shimbo et al. showed that synthetic miR-143 was transduced into YJP-1 macrophage cells and that the secreted miR-143-exosomes could be efficiently transferred to osteosarcoma cells—which resulted in suppression of the migration of osteosarcoma cells [76]. miR-146b was transfected into MSCs; MSC-derived exosomes expressing miR-146b inhibited cancer growth [56]. Ohono et al. demonstrated that engineered exosomes containing large amounts of let-7a were able to facilitate tumor suppressions [39]. However, there is an open question concerning whether it is possible to control the selective sorting of antitumor drugs into multivesicular bodies. Further approaches to establish efficient cargo-loading strategies to yield exosomes enriched in therapeutic drugs or oligonucleotides are required.

### 4.4. Exosome-Based Immunotherapy

In general, exosomes possess specific molecules that reflect their cellular origins. Therefore, exosomes derived from antigen-presenting cells (APCs) such as B cells and dendritic cells (DCs) can play an important role in anti-tumor response in immune stimulation and regulation [77]. In cancer immunity, DCs are involved in the first step of immune responses aimed at eliminating tumor cells through triggering of tumor-specific cytotoxic lymphocytes [77]. Also, exosomes from DCs exhibit surface expression of functional MHC-peptide complexes, costimulatory molecules and other components that interact with immune cells [78]. Indeed, several reports have shown that exosomes derived from DCs activate T-cell and NK-cells; several clinical studies using exosomes derived from DCs have been attempted against advanced cancer due to their high potential and benefits for immunotherapy [77,78,79]. Besse et al. conducted a Phase I study using dendritic cell-derived exosomes (Dex) against end-stage non-small cell lung cancer. Twenty-two patients received IFN-γ-boosted Dex and 7 (32%) of them experienced stable disease for more than 4 months, with the elevation of NK cell function observed in these patients [80]. Tumor-derived exosomes carry MHC-I molecules, heat shock protein 70 (HSP70) and antigens, all of which work as source of specific stimuli for the immune response against cancer, thus, can be the potential candidate to be used as cell-free tumor vaccines [77]. Rao et al. demonstrated that hepatocellular carcinoma cell-derived exosomes could elicit a stronger DC-mediated immune response following the increased numbers of T lymphocytes and elevated levels of IFN-γ in the tumor microenvironment [81]. Ascites-derived exosomes of advanced colorectal carcinoma (CRC) patients were used to treat 40 advanced CRC patients with exosomes alone or exosomes plus granulocyte-macrophage colony-stimulating factor (GM-CSF) [82]. Although any significant therapeutic effects were not observed when only exosomes were used for treatment, exosomes plus GM-CSF activated CD8+ cytotoxic T-lymphocyte, eliciting an antitumor immune response; this suggested that exosomes have the potential to become cancer immunotherapeutic options in the future.

Recently, Cheng et al. revealed that melanoma cells released programmed cell death-ligand 1 (PD-L1)-positive exosomes into the tumor microenvironment and circulation to systemically battle anti-tumor immunity. Therefore, they suggested that disrupting the interaction between exosomal PD-L1 and T-cell PD-1 could be a novel therapeutic strategy [83].

## 5. Conclusions

Ovarian cancer-derived exosomes play a crucial role in tumor progression. These small vesicles work as potent signaling molecules between cancer cells and the surrounding cells that compose the tumor microenvironment. Exosomes reeducate their microenvironments and promote infiltration, pre-metastatic niche formation and peritoneal dissemination in ovarian cancer. Therefore, further elucidation of the relationship between ovarian cancer and exosomes is essential to comprehend the mechanism of tumor progression; this would lead to the development of novel therapeutic methods. Additionally, exosomes may be promising options for cancer therapy, including ovarian cancer. The potential of exosome-based therapies, including novel drug delivery vehicles, biomolecular therapy and antitumor immune therapy due to their pathological features, stability in body fluids and specific tissue targeting have been demonstrated. However, there remain challenging issues in the used of exosomes as disease therapeutics. These include the limited information on stable isolation, purification, storage and administration of targeted exosomes. Besides, exosome engineering should overcome the challenges of the low loading efficacy associated with the exogenous method and the isolation difficulty associated with the endogenous method. Further, it is not clear how engineered exosomes affect ovarian cancer cell behavior, considering the complexity of their content. As such, further studies are required to evaluate the use of exosome-based therapies as clinical interventions.

Currently, ClinicalTrialis.gov (https://clinicaltrials.gov/) has disclosed 77 clinical trials involving exosomes. For ovarian cancer, three clinical trials of exosomal biomarkers were on-going at the end of 2019. However, to date, there are no clinical trials concerning exosome-based therapy in ovarian cancer. Given that ovarian cancer is associated with EVs, there are high expectations about EV-based therapies as novel clinical strategies to overcome this intractable disease.

## Figures and Tables

**Figure 1 cells-09-00814-f001:**
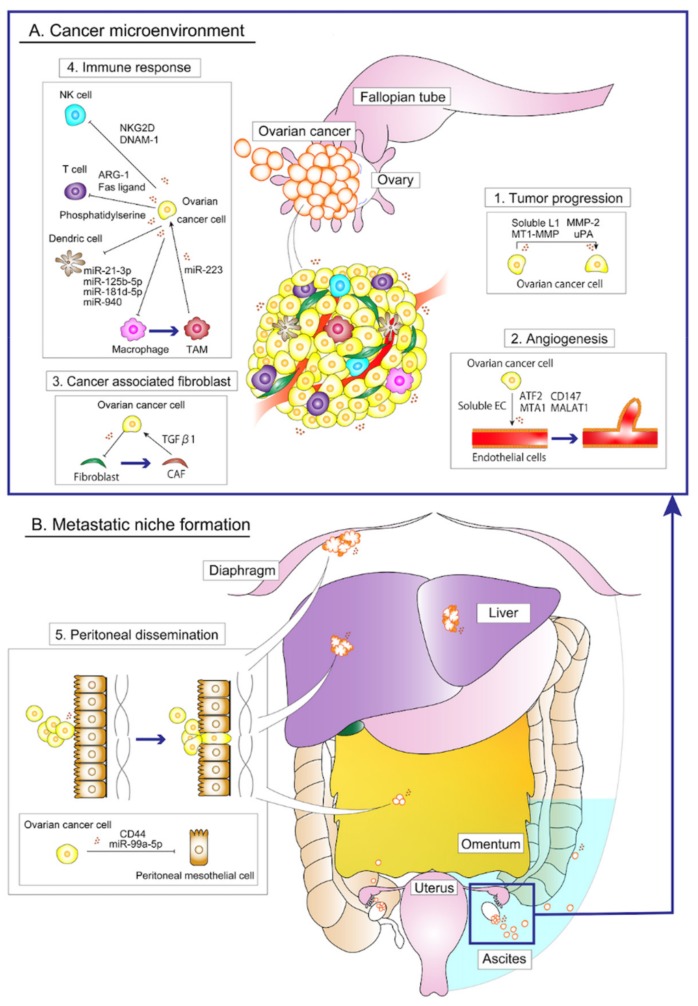
Overview of the roles of ovarian cancer-derived exosomes during local progression and peritoneal dissemination. (**A**) The relationship between cancer-derived exosomes and the cancer microenvironment is summarized. (1) Ovarian cancer cell-derived exosomes and those isolated from ascites promote the progression of ovarian cancer cells [35,36]. (2) Ovarian cancer cell-derived exosomes loaded biomolecules, such as ATF2, MTA1, CD147 and MALAT1, are transferred into endothelial cells where they induce angiogenesis [31,32,33,50]. (3) Ovarian cancer cell-derived exosomes convert fibroblasts into cancer associated fibroblasts (CAFs). Thereafter, CAF-derived exosomes promote epithelial-mesenchymal transition to ovarian cancer cells [48,49]. (4) Ovarian cancer cell-derived exosomes inhibit immune cells and facilitates the conversion of macrophages to tumor-associated macrophages [38,40,41,42,44,45]. (**B**) The involvement of exosomes in peritoneal dissemination. (5) Ovarian cancer-derived exosomes induce apoptosis of peritoneal mesenchymal cells and enhance the infiltration of ovarian cancer, thereby promoting the formation of the metastatic niche [21,45,46]. “→” thin arrows mean stimulating effects and “T” symbols mean suppressive effects.

**Table 1 cells-09-00814-t001:** Biomolecules included in exosomes and their role in the progression of ovarian cancer.

Molecules	Type of Molecules	Recipient Cell	Role	Reference
ATF2, MTA1, ROCK1/2	Protein	HUVEC	Angiogenesis	[31]
MALATI	Protein	HUVEC	Angiogenesis and peritoneal dissemination	[32]
sE-cad	Protein	HUVEC	Angiogenesis and peritoneal dissemination	[33]
MTI-MMP, MMP-2, MMP-9, uPA	Protein	Cancer cell	Invasion	[34]
CD24, EpCAM	Protein	Cancer cell	Invasion	[35]
soluble L1(CD171)	Protein	Cancer cell	Migration	[36]
TGF-β	Protein	Cancer cell	EMT and migration	[37]
miR223	miRNA	Cancer cell	Chemoresistance	[38]
let7a-f, miR-200a-c	miRNA		Local invasion and metastasis	[39]
NKG2D, DNAM-1 ligand	Protein	NK cells	Immunosuppression	[40]
ARG-1	Protein	T-cell	Immunosuppression	[41]
Phosphatidylserine	Phospholipid	T-cell	Immunosuppression	[42]
Fas ligand	Protein	T-cell	Immunosuppression	[43]
miR21-3p, miR125b-5p, miR181d-5p	mRNA	Macrophage	Proliferation/migration (M2 polarization of macrophages)	[44]
miR-940, miR-222-3p	miRNA	TAMs	M2 phenotype polarization, Proliferation and migration	[45]
MMP1 mRNAs	mRNA	MeT-5 HPMC	Destruction of the peritoneal mesothelial barrier	[21]
miR-99a-5p	miRNA	HPMCs	Destruction of the peritoneal mesothelium barrier	[46]
CD44	Protein	HPMCs	Tumor cell invasion for peritoneal dissemination	[47]

Abbreviations: ATF2, activating transcription factor 2; MTA1, metastasis associated 1; ROCK1/2, rho-associated kinase 1/2; sE-cad, soluble E cadherin; MALAT1, metastasis associated in lung adenocarcinoma transcript-1; ARG-1, arginase-1; NKG2D, natural killer group 2, member D; DNAM-1, DNAX accessory molecule-1; EpCAM, epithelial cell adhesion molecule; MMP-2, matrix metalloproteinase-2; MMP-9, matrix metalloproteinase-9; uPA, urokinase-type plasminogen activator; MT1, metallothionein 1; TGF-β, transforming growth factor; HUVEC, human umbilical vein endothelial cell; NK cell, natural killer cell; PBMC, peritoneal blood mononuclear cell; TAM, tumor-associated macrophage; HPMC, human peritoneal mesothelial cell.

**Table 2 cells-09-00814-t002:** Summary of the potential molecules, loading approaches and roles in exosome-based therapies.

Source of EVs	Loading Approach	Cargo	Type of Cancer	Key Molecules	Role	Reference
LNCaP PC-3	Incubation	Ptx	Prostate cancer		Enhance the cytotoxic effect	[58]
Macrophage	Sonication	Ptx, Dox	Lung carcinoma		Drug-loaded exosomes indicated the efficacy for MDR and suppressed metastasis	[59]
DCs	Electroporation	Dox	Breast cancer		Loaded exosomes delivered Dox specifically to tumor tissues, leading to inhibition of tumor growth without overt toxicity	[60]
HeLa HTB-177 Plasma	Electroporation	siRNA	Uterine cervical cancer	MAPK-1	Vesicles effectively delivered the administered siRNA into monocytes and lymphocytes	[61]
Fibroblast	Electroporation	shRNA, siRNA	Pancreatic cancer	KRAS	Suppress pancreatic cancer progression in mouse models	[54]
Fibroblast MSCs	Electroporation	siRNA	Pancreatic cancer	KRAS	Suppress pancreatic cancer progression and metastases in mouse models	[62]
HEK293T MCF-7	Sonication	siRNA	Breast cancer	HER2	Suppress breast cancer in vitro and vivo	[63]
THP-1	Incubation Electroporation	miRNA		BCL-2	Improve miRNA transfection	[64]
Plasma	Electroporation	miRNA	Liver cancer	BCL2α CASP3 RAB14	Promote the apoptosis of hepatocellular carcinoma cells	[65]

Abbreviations: DC, dendritic cells; MSCs, mesenchymal stem cells; TC, T-cells; Ptx, Paclitaxel; Dox, doxorubicin; siRNA, small interfering RNA; miRNA, micro RNA; MDR, multidrug resistance.

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
