# Peer review of "Pathophysiological Role and Potential Therapeutic Exploitation of Exosomes in Ovarian Cancer"

_cells, 2020, doi:10.3390/cells9040814_

Round 1

Reviewer 1 Report

Shimizu and colleagues described in review: Role and therapeutic potential of extracellular 3 vesicles in ovarian cancer, the role namely of exosomes in treatment and development of ovarian cancer. The paper is well written and organized. I  recommend acceptance in present form.  

Author Response

We appreciate the kind comment from reviewer 1.

Reviewer 2 Report

In this review manuscript, Shimizu et al. have summarized the current development related to exomes in ovarian cancer (OC) progression and their therapeutic potential.

Overall, the organization of this review is clear; it brings together some current knowledge and contributes to the current literature. Nonetheless, there are some areas that need improvement.

While the authors have discussed some key refs like those in 4.2.1 and other sections. These could be expanded a little more. For instance, were specific pathways affected? Are there any connections among these articles? Or by bring these refs together here do the Shimizu et al. try to convey certain concept?

Although the authors have provided some limitations largely at the end, more is needed. For example, will the complex content of exosomes be a potential issue in developing exosome-based therapy? Will these complex molecules all contribute to OC therapy? With this in mind, will exosome as a carrier of cancer therapeutic drugs be superior to lipid-based nanoparticle? miRNAs are unlikely target-specific, i,e, they affect numerous targets, will this be a limitation?

How did the reference articles were selected for review here? Was the selection based on PRISMA Guidelines?

As this review focuses on exosomes, it will be better to use “exosomes” in the tile instead of “extracellular vesicles”.

Be consistent about the size of apoptotic bodies, 1-5μm or 800-5000 nm?

The sentence “Although … … improved [12 -14]”, lines 52-55 is not clear.

Line 150, HPMC is cell.

It has been used several times: OC exosomes transform fibroblasts. Did the authors mean these fibroblasts become cancer cells? Please rephrase it.

How can exosome deliver drugs specifically to tumor?

Fig 1A – Please improve the fig, do all arrows mean stimulation?

Section 4.2.3. – please clarify: preclinical trials and translational clinical trials?

Author Response

Reviewer 2

In this review manuscript, Shimizu et al. have summarized the current development related to exomes in ovarian cancer (OC) progression and their therapeutic potential.

Overall, the organization of this review is clear; it brings together some current knowledge and contributes to the current literature. Nonetheless, there are some areas that need improvement.

While the authors have discussed some key refs like those in 4.2.1 and other sections. These could be expanded a little more. For instance, were specific pathways affected? Are there any connections among these articles? Or by bring these refs together here do the Shimizu et al. try to convey certain concept?

We added the following sentences in 4.2.1.

“These data strongly suggest that exosome-encapsulated chemotherapeutic drugs can exert robust cytotoxic effects to cancer cells without increasing apparent adverse events, indicating they can serve as a promising novel delivery platform for the treatment of ovarian cancer which is resistant to the standard chemotherapy.”

Although the authors have provided some limitations largely at the end, more is needed. For example, will the complex content of exosomes be a potential issue in developing exosome-based therapy? Will these complex molecules all contribute to OC therapy? With this in mind, will exosome as a carrier of cancer therapeutic drugs be superior to lipid-based nanoparticle? 

To address these comments, we added the following sentences in 4.1 and 5, respectively.

“Liposomes and polymeric nanoparticles have been the most common engineered systems for DDS for cancer therapy. Compared with these nanovectors, exosomes can avoid the endosomal and lysosomal pathways and deliver their cargoes directly into cytoplasm of recipient cells [53].”

“Further, it is not clear how engineered exosomes affect ovarian cancer cell behavior, considering the complexity of their content.”

miRNAs are unlikely target-specific, i,e, they affect numerous targets, will this be a limitation?

As suggested, we added the following sentence in 4.2.3.

“Further, since each miRNA regulates hundreds of gene expression, therapeutic miRNAs might induce unwanted effects when they are introduced into normal cells [69].”

How did the reference articles were selected for review here? Was the selection based on PRISMA Guidelines?

Although this article is not a systematic review, we carefully performed the bibliographic search according to the following protocol.

The bibliographic search for this review was conducted until October 2019, with limits for the English language. A combination of the following medical subject headings or keywords was included: “exosome(s)”, “extracellular vesicles”, “nanoparticle”, “ovarian cancer”, “Drug delivery systems”, “miRNA”, “siRNA”, “chemotherapeutic drug”, and “immunotherapy”.

We searched Medline (through PubMed), the Institute for Scientific Information Web of Science database, and websites for the registration of controlled trials. The bibliographies of retrieved articles, books, and expert opinion review articles were manually searched.

As this review focuses on exosomes, it will be better to use “exosomes” in the tile instead of “extracellular vesicles”.

Since Reviewer 3 also suggested the change of the title, we changed the title as follows:

“Pathophysiological role and potential therapeutic exploitation of exosomes in ovarian cancer”

Be consistent about the size of apoptotic bodies, 1-5μm or 800-5000 nm?

We unified the size of apoptotic bodies to1-5 µm.

The sentence “Although … … improved [12 -14]”, lines 52-55 is not clear.

We revised the sentence as follows.

“the clinical cure rate in patients with advanced-stage ovarian cancer has not substantially improved” 

Line 150, HPMC is cell.

We appreciate this comment. We revised it.

It has been used several times: OC exosomes transform fibroblasts. Did the authors mean these fibroblasts become cancer cells? Please rephrase it.We rephrase “transform” to “convert”. 

How can exosome deliver drugs specifically to tumor?

To address this comment, we added the following sentence in 4.1.

“Previous studies have shown that exosomes are specifically up taken by cancer cells rather than normal cells through endocytosis [54], although the underlying mechanisms have not been completely understood. Goh, et al. reported that exosomes present the particular tetraspanins and integrins on their surface, which lead to fulfil the important role of uptake and targeting ability [55].”

Fig 1A – Please improve the fig, do all arrows mean stimulation?

We revised the figure and added the following explanation in the figure legend.

““→” thin arrows mean stimulating effects, and “T” symbols mean suppressive effects.” 

Section 4.2.3. – please clarify: preclinical trials and translational clinical trials?

As suggested, we revised this sentence as follows: “Although several miRNA therapies have been attempted in preclinical trials and many promising results have been reported, the outcomes of a few translational clinical trials for miRNA therapy have been disappointing [70].”

Reviewer 3 Report

This is a very interesting, comprehensive and timing review on a topic of interest for Cells' readers. Tables and iconography are clear and informative. I have enjoyed the reading. The article is well written and organized. Still, authos may wisg to improve the presentation taking into consideration the followings suggestions:

  1. the title could be more specific substituting 'extracellular vesicles' with 'exosomes', since this is at the end the type of EV the authors are dealing with.
    2.Title: I suggest "Pathophysiological role and potential therapeutic exploitation of exosomes in ovarian cancer". Otherwise, it seems that the 'natural' exosomes have the therapeutic potential.
    3. consider to anticipate the paraghraph 3.1 (facultative)
    4. MAJOR: there is no mentioning of inflammatory cytokines within exosomes and their role in OC progression. Authors should consider the cytokine-containing exosomes from CAFs and found in ascites. This information must be included.
    MINOR: please, double check the grammar and typos. Below some examples:
    - line 12: not sure that 'indispensable' is appropriate;
    - line 55: has NOT substantially improved;
    - line 90: signal, preserves... (insert comma and delete 'and')
  2. REFERENCES: consider refs on cytokines and exosomes.

Author Response

Reviewer 3

This is a very interesting, comprehensive and timing review on a topic of interest for Cells' readers. Tables and iconography are clear and informative. I have enjoyed the reading. The article is well written and organized. Still, authors may wish to improve the presentation taking into consideration the followings suggestions:

the title could be more specific substituting 'extracellular vesicles' with 'exosomes', since this is at the end the type of EV the authors are dealing with.

2.Title: I suggest "Pathophysiological role and potential therapeutic exploitation of exosomes in ovarian cancer". Otherwise, it seems that the 'natural' exosomes have the therapeutic potential.

We revised the title as advised.

3. consider to anticipate the paragraph 3.1 (facultative)
4. MAJOR: there is no mentioning of inflammatory cytokines within exosomes and their role in OC progression. Authors should consider the cytokine-containing exosomes from CAFs and found in ascites. This information must be included. 

To address this comment, the following sentences were added in 3.2.

“CAF-derived exosomes contain TGFβ1, and exosomal TGFβ1 enhanced the migration and invasion ability of ovarian cancer cells and the promotion of epithelial-mesenchymal transition (EMT) by activating the SMAD signaling [32].”

“Ovarian cancer derived exosomes can enhance the production of IL-6 in monocytes through toll-like receptor (TLR) activation. Thereafter, IL-6 activates the signal transducer and activator of transcription 3 (STAT3) pathway in immune cells, stromal cells, and tumor cells, which supports immune escape of cancer cells [33].”

MINOR: please, double check the grammar and typos. Below some examples: 
- line 12: not sure that 'indispensable' is appropriate. 

We rephrased “indispensable” to “essential”.

- line 55: has NOT substantially improved;

We appreciate this comment. We revised it.

- line 90: signal, preserves... (insert comma and delete 'and')

We revised the corresponding section as suggested.

REFERENCES: consider refs on cytokines and exosomes

We added the contents of reference 33 which focused on cytokines and exosomes.

Round 2

Reviewer 2 Report

My comments have been addressed to my satisfaction in general. The revision is an improved manuscript; I supports its publication.